# Health Service Accessibility, Mental Health, and Changes in Behavior during the COVID-19 Pandemic: A Qualitative Study of Older Adults

**DOI:** 10.3390/ijerph19074277

**Published:** 2022-04-02

**Authors:** Sofia von Humboldt, Gail Low, Isabel Leal

**Affiliations:** 1William James Center for Research, ISPA—Instituto Universitário, 1149-041 Lisbon, Portugal; ileal@ispa.pt; 2Faculty of Nursing, University of Alberta, Edmonton, AB T6G 2R3, Canada; gaill@ualberta.ca

**Keywords:** access to health services, changes in behavior, health service accessibility, mental health, older adults

## Abstract

The COVID-19 pandemic has affected the access of older adults to health services. The two objectives of this study are understanding the influence of the COVID-19 pandemic on older adults’ access to health services and exploring how health service accessibility during the pandemic influenced older adults’ mental health and self-reported changes in behavior. This study included 346 older adults. Content analysis produced five themes: (1) decreased physical accessibility to health care providers (78%); (2) increased use of online health services and other virtual health care (69%); (3) growth in the online prescription of medication (67%); (4) difficulty obtaining information and accessing non-communicable disease and mental health indicators (65%); and (5) postponement of medical specialist consultations (51%). Regarding mental health, three themes emerged: (1) increased symptoms of anxiety, distress, and depression (89%); (2) the experience of traumatic situations (61%); and (3) the augmented use of alcohol or drugs (56%). Finally, the following changes in behavior were indicated: (1) frustrated behavior (92%); (2) emotional explosions (79%); and (3) changes in sleeping and eating behaviors (43%). Access to health services may have influenced the mental health and behavior of older adults, hence interventions in a pandemic must address their interactions with health services, their needs, and their well-being.

## 1. Introduction

The COVID-19 pandemic has affected individuals worldwide, becoming more than just a medical manifestation. It has produced anxiety, disruption, stigma, and stress [1]. A study by the World Health Organization [2] that included 105 countries indicated that the COVID-19 pandemic affected all health services (e.g., mental health; essential services for communicable and non-communicable diseases; and neonatal, maternal, reproductive, child, and adolescent health and nutrition).

Older adults are more prone to social isolation, financial difficulties, domestic violence, and abuse; difficulties in accessing care; and increased anxiety over becoming infected with COVID-19 [1]. To date, insufficient data on COVID-19 and older adults have been shared. In this context, more than 95% of deaths due to COVID-19 worldwide were people aged 65 years and older, and more than half of deaths were people over 80 [3]. As of 30 December 2021, among older adults, the total number of cases of COVID-19 had reached about 5.34 million, 56.6% of whom were aged between 65 and 74 years, 28.4% were between 75 and 84 years old, and the remaining 15% were over 85 years old [3]. As of early May 2021, 82% of individuals aged 65 years or older had already received at least one dose of the vaccine [4].

Relevant challenges have called into question some health services, including those in mental health areas [5]. The burden and neglect of psychiatric services both during and after this health crisis have been a major risk [6]. In addition, mental health services are one of the most sought after due to measures for social distancing and the suffering felt during the COVID-19 pandemic. Some in-person health services were replaced by online services, namely routine consultations and procedures and also prescriptions, while in-person interactions were replaced by online interactions, or were postponed, or canceled [6]. In relation to this, technological tools proved to be relevant for the assessing of older individuals in health services [7,8].

Feelings of depression and anxiety may be heightened due to fear of COVID-19-related morbidity and mortality in older adults [9]. Often, anxiety levels among older individuals are marked by health concerns. With the current pandemic, these concerns can be compounded and all of the associated policies (such as social distancing) must also play a role in the compounding of such concerns. Health-related concerns about COVID-19 and ageism during this pandemic have been associated with greater symptoms of anxiety in older adults. In addition, symptoms of anxiety, health concerns, and changes in behavior (e.g., emotional outbursts, fear, and frustration manifestations) related to the virus were higher among older people [10]. In fact, almost 25% of adults aged 65 and over reported anxiety or depression in August 2020; for individuals aged 80 or over, the rate was 20% [11].

Healthy aging implies a balance between security, social well-being, and mental health [12,13,14,15]. The COVID-19 pandemic challenged the concept of healthy aging. Clinically stable older people with psychiatric disorders were advised to suspend their monthly psychiatric clinic visits for obtaining maintenance medications [16].

Moreover, activities that involved socialization and interaction with friends, family, and other acquaintances, as well as physical exercise in groups, were discouraged in order to avoid the spread of the virus. This was particularly evident in communities with poor conditions [17].

The transmission rate of the COVID-19 virus is high and older adults show high mortality rates [18]. Moreover, infection can affect psychological well-being [19].

The recommendations of the WHO [20] for reducing the risk of disease transmission have affected social interaction. These measures were necessary to control social distancing and to reorganize activities that are important for quality of life [12]. Moreover, due to social distancing, routine procedures and consultations have frequently had to be canceled [21].

Social interaction is a crucial element for older populations, including for reducing levels of depression and stress, promoting positive cognitive effects, and for helping the well-being of the older community [22].

During the COVID pandemic, compromised transport capacity prevented older people from accessing places to meet their basic needs and their health care providers [16]. Moreover, mass quarantines and restrictions have become a challenge for older populations in accessing the treatments they need. Indeed, a large part of the older population had difficulties in accessing medication and receiving the routine medical care they needed during the pandemic [16].

To date, there have been few studies exploring older adults’ access to health services during the COVID-19 pandemic and how such access is linked to changes in mental health and behavior among older adults. As such, the two main objectives of this study were: (a) to better understand the COVID-19 pandemic’s influence on older adults’ access to health services and (b) to explore how health service accessibility during the COVID pandemic influenced the mental health of older adults and their self-reported changes in behavior.

## 2. Materials and Methods

### 2.1. Recruitment and Sampling

For this study, 373 people were initially contacted, and 367 agreed to participate. Recruiting methods were varied, namely through message boards, senior universities, community center list-serves, personal emails, and social network ads. Twenty-one participants were excluded due to lack of availability or incomplete data provided, yielding an unweighted response rate of 92.8% and a weighted response rate of 93.8%.

The final purposeful sample included 346 participants from three nationalities (Brazilian, English, Portuguese), (64.2% women), aged between 65 and 79 years (M = 71.5; SD = 4.65). In total, 77.7% of participants were married or in a romantic relationship, 12.1% lived alone, and 17.1% had finished university (see Table 1 and Appendix A Table A1).

This qualitative study was developed through semi-structured interviews with older adults. Initially, the objective of the study was explained to the participants, including the information that all information would be used only for this study and would be anonymous. Participants responded to an online questionnaire (e.g., Skype, Survey Monkey, Zoom, and Whatsapp) with their informed consent. Participants were provided with telephone or online support at their disposal to help with any queries. To build the sample for this study, participants had to meet the following inclusion criteria: (1) age 65 years or older; (2) possess a clear understanding of the decision to participate in the study; (3) no history of problems with impairment of cognitive functions, psychiatric or neurological illnesses, or history of drug or alcohol abuse, among others; and (4) possess minimal knowledge about the use of new technologies (smartphones, tablets, computer, apps, etc.).

In order to understand the impact of the pandemic on the accessibility of health services to the older population, semi-structured and qualitative interviews were conducted with individuals who attended life-long learning and community centers. Individual interviews were scheduled, taking into account the availability of the participant and the interviewer, with a duration of approximately 30 min. Between 1 May and 30 July 2021, interviews were carried out and data collected. All interviews were analyzed and transcribed in full, and thematic matrices were later built with the most relevant information for the study. All procedures were approved by the Ethics Committee of the William James Center for Research, ISPA–Instituto Universitário, and were in accordance with the ethical standards of the 1964 Helsinki Declaration and its subsequent amendments or with comparable ethical standards.

### 2.2. Data Analysis

After obtaining the sample information provided in the semi-structured interview protocol, a detailed analysis was performed to ensure exhaustive contact with the data. Data were analyzed by applying content analysis [23].

First, for a more organized and complete analysis of the information, a codebook was created wherein each important category was assigned a code [23]. All interviews were coded by three researchers independently, and all processes were considered reliable (k = 0.87). In this study, a *p*-value ≤ 0.05 was considered significant in all analyses.

The main themes and sub-themes were defined and then the categorization process began. At this stage, with the consent of the three psychologists, the main themes were grouped into clear and independent categories. These categories were given short and intuitive names. In order to have a reliable and valid system, general principles of the classification and categorization of qualitative data were taken into account: homogeneity (they were organized based on common elements), relevance (reflects the importance of the categories), and objectivity and fidelity (the categories were objective and reliable). These principles were maintained until the end of the process. The above categorization process was handled manually.

Finally, the results interpretation matrix was created to carry out the theoretical and empirical discussion of the obtained data. This phase had two stages: (1) quantitative descriptive analysis, including calculations of means, medians and percentages, and frequencies and means of sociodemographic variables and (2) qualitative analysis of the information that emerged from the relationship between the theoretical models used and the empirical reality.

## 3. Results

In this study, five main themes for the first objective were found, namely (1) a decrease in the physical accessibility of health care providers (78%); (2) increased use of online health services and other virtual health care (69%); (3) the growth of online medication prescription (67%); (4) difficulty in obtaining information and accessing non-communicable disease and mental health indicators (65%); and (5) the postponement of medical specialist consultations (51%). Key quotes representing the diversity of the narratives of older people were selected to illustrate the diversity of the studied sample. All names are pseudonyms.

### 3.1. Theme 1: Decrease in Physical Accessibility of Health-Care Providers

A good proportion (78%) of participants mentioned that during the pandemic they could not make face-to-face appointments with the family doctor since this implied using public transport or being in a public place for the consultation and thus being more exposed to the virus. Dilara mentioned that “with the increase in COVID-19 cases, our consultations had to be carried out virtually” (Dilara, female, 87 years old). Hugo also reported that “initially it was difficult to adapt to the technological way, but due to social distancing and all other restrictions, it was necessary that it happen this way” (Hugo, male, 76 years old).

### 3.2. Theme 2: Increased Use of Online Health Services and Other Virtual Health Care

Participants (69%) also verbalized that over time it became easier to access the consultations virtually as they were helped by family members. Briana reported that “my son, or sometimes even my grandson, would help me make the online appointment. Then I learned and it became easier” (Briana, female, 78 years old). This online mode of consultation was promoted by the health services and over time patients got used to it, and like Sara, they said that “I sometimes even preferred the consultations to be online because that way I didn’t have to move” (Sara, female, 91 years old).

### 3.3. Theme 3: Growth of Online Medication Prescription

When asked about online of medication prescription, 67% of the sample stated that COVID-19 had had an impact on medication prescription. “Now everything is done online: the prescription is a virtual document, it is possible to order the medication, and even ask for it to be delivered to the patients’ homes,” Filip reported. “Thanks to the pandemic, we are exploring other technological aspects reaching the prescription of medications” (Filip, male, 82 years old). Among other positive factors, Sean stated that “now everything seems easier. Everything is easier, faster, and better” (Sean, male, 77 years old).

### 3.4. Theme 4: Difficulty in Obtaining Information and Accessing Non-Communicable Disease and Mental Health Indicators

Several participants (65%) verbalized the difficulty and disorientation in obtaining information and controlling indicators of chronic diseases as a theme influenced by COVID-19. This is because a large portion of older people had not had consultations for disease control, such as cancer, for example. William verbalized that “Age doesn’t help when we have chronic problems. Due to this health crisis, many health workers were made unavailable as they were trying to prevent cases from continuing to grow, but what about the patients with chronic illnesses? Many people still need help and it is hard controlling chronic disease indicators” (William, male, 68 years old). Diana also reported, “although we have included many technological aspects due to the pandemic, many of us older people still have difficulty getting information. We had to adapt in a frantic way, which is difficult at our age” (Diana, female, 85 years old).

### 3.5. Theme 5: Postponement of Medical Specialist Consultations

The last category of this study was the postponement of specialist appointments, such as in the area of cancer, vascular surgery, and mental problems, with this being mentioned by 51% of participants. Diego mentioned that “maybe the pandemic helped to make some situations easier and faster, but when it comes to specialist consultations, this is not true. I think many doctors had to go to help in hospitals and health centers to control cases of COVID-19, preventing them from carrying out their appointments, which is why they were postponed” (Diego, male, 76 years old). It was also verbalized by Karoline that “due to social distance, disinfection of offices, and other restrictions, many appointments had to be rescheduled due to the time needed to get everything in good condition” (Karoline, female, 73 years old).

Three main themes emerged from the narratives regarding the second study objective: (1) increased symptoms of anxiety, distress, and depression (89%); (2) the experience of traumatic situations (61%); and (2) the augmented use of alcohol or drugs (56%).

### 3.6. Theme 1: Increased Symptoms of Anxiety, Distress, and Depression

The most commonly reported theme (89%) was increased symptoms of anxiety, distress, and depression. For these participants, the experience accessing online health services was perceived as negative and a major source of distress, fear, frustration, anxiety, and depression. “A lot has changed, and I take longer to get used to these changes. Now it’s all through links and digital platforms that many of us don’t know about. Sometimes I get anxious working with new things, and when I can’t handle it, I get really nervous, anxious, and sometimes very sad.” (Dino, male, 71 years old).

### 3.7. Theme 2: Experience of Traumatic Situations

The second most verbalized theme was the experience of traumatic situations (61%). “We hear in the news that cases of COVID-19 continue to increase and that several people need medical care, but we never realized its proportion. I was in the hospital once waiting for a cancer exam and it was horrible to see the amount of people who were waiting to be seen by a doctor. In the end, I was sent home and had to wait two months for that exam to be performed since the medical staff were not available for the other diseases. I was not well and I couldn’t deal with that. I had nightmares and I thought I was going to die. It was traumatic for me.” (Charlotte, female, 66 years old).

### 3.8. Theme 3: Augmented Use of Alcohol or Drugs

More than half (56%) of the participants indicated an increase of their alcohol and drug consumption. Many reported that taking substances helped them forget their irritation, fear, and frustration with health accessibility: “Nothing seems to get better. We’ve been dealing with this for over one year now and despite all the vaccinations, online consultations, and restrictions, it seems to have no end. Sometimes what helps me not to think about it is to drink and I drink too much.” (Daniel, male, 79 years old).

Finally, three main changes in behavior were reported by participants: (1) frustrated behavior (92%); (2) emotional explosions (79%); and (3) changes in their sleeping and eating habits (43%).

### 3.9. Theme 1: Frustrated Behavior

The great majority of the participants felt frustrated when accessing health services during the COVID-19 pandemic: “I’m not a big fan of new technologies, but I like online consultations… it made my life easier. The problem is that I don’t understand how it works. It’s frustrating to keep asking my children or my neighbors for help, but I just can’t understand how it works.” (Carol, female, 68 years old).

### 3.10. Theme 2: Emotional Explosions

Most participants (79%) reported experiencing emotional explosions during the process of accessing health services during the pandemic: “I feel like I’m a more emotional person now; how could I not be when you’re in a waiting room and you look around and see children, youths, parents and other older people like me who need medical attention. Sometimes I feel like crying right then and there. I don’t cry there because I get embarrassed, but it stays in my mind for the rest of the day. And when I get home, I just melt down and cry and cry for hours. I don’t know what to do with it.”

### 3.11. Theme 3: Changes in Their Sleeping and Eating Habits

Nearly half (43%) of participants experienced changes in behavior concerning their sleeping and eating habits. “I had to wake up early to get ready, take public transport, wait in the hospital, and then I had my appointment. Since the pandemic started, I sleep less well because I get very nervous about the fact that I do not know when I can get a medical appointment, plus I do not feel like eating. I get so anxious I do not feel like it”.

## 4. Discussion

This study aimed to understand the influence of the COVID-19 pandemic on older adult access to health services, and to explore how health service accessibility during the COVID-19 pandemic influenced the mental health of older adults and their self-reported changes in behavior.

For the first objective, the participants perceived a decrease in the physical accessibility of health-care providers, an increased use of online health services and other virtual health care, a growth of online medication prescription, a difficulty in obtaining information and accessing non-communicable disease and mental health indicators, and a postponement in medical specialist consultations.

The decrease in the physical accessibility of health-care providers was referred to by 78% of the participants. The pandemic and its associated restrictions created obstacles during this phase, where the patient–doctor relationship was affected, both in terms of communication and, inevitably, in terms of trust. More than 75% of the participants referred to difficulties in communicating with doctors and complained about the difficulty in accessing health services. Physical access is relevant as it may affect the quality of care and willingness of patients to engage in preventive care [24]. In addition, the doctor–patient interaction may be negatively affected by online meetings by giving older patients the possibility of contacting the doctor whenever necessary, thus affecting the quality of the service provided [25].

In fact, the lack of availability of consultations for acute and severe pathologies decreases the possibility of timely treatment, with this potentially impacting morbidity and mortality. During the pandemic, an increase in mental illnesses was observed due to isolation, which was not supported by a greater availability of support services. Moreover, there were reports of difficulties in accessing mental health specialists [26].

The increase use of online health services and other virtual health care was the second most reported theme. Telehealth ended up facilitating access to health care for some patients; however, for others, who do not have access to adequate technology, this access was reduced. We are currently witnessing potential concerns about patient involvement in telecommuting consultations and the value of health care provided through telehealth. Furthermore, in order to guarantee equity in all communities, it is necessary to pay special attention to facilitating access to telehealth [27].

With the COVID-19 pandemic, telemedicine has gained greater importance. Telephone or video-call consultations have been models used between patients and physicians to ensure safety [28]. The types of consultations for which a video encounter could replace an in-person visit include chronic disease reviews, counselling or other talking therapy, administrative appointments (e.g., for sick notes), some medication reviews, and triage when telephone visits are insufficient [29]. Also, video consultations are appropriate for people with medical conditions such as heart failure, chronic obstructive pulmonary disease, hypertension, asthma, and diabetes [30].

The number of older adults with experience with technology showed an increase before the beginning of the pandemic. In fact, between 2013 and 2017, mobile phone ownership increased by almost 25% among older individuals, and currently around three quarters of the older population goes online every day [31,32].

Older adults also reported being motivated to learn how new technologies work and are excited to use these new techniques and skills when necessary [32]. However, some older adults reported difficulties using online consultation systems. Older patients may resist using technology to make online consultations due to lack of familiarity with technology and lack of knowledge.

There are some authors who state that their visions for the future consist of chronic disease clinics where consultations are interspersed between virtual and ad hoc face-to-face consultations when necessary. Telemedicine has already proven to be a tool that facilitates contact with the health team, reduces mortality, and induces a better quality of life in the management of chronic diseases [33,34].

The online prescription of medication has been indicated as a frequent practice for older adults. The literature reiterates the advantages of online medication prescription, such as being convenient and efficient, in particular for population groups which show difficulties in health access [29].

Participants indicated difficulties in obtaining information and accessing non-communicable disease and mental health indicators. In this regard, caregiver burden may have increased due to reduced availability of support [35]. Health units were overwhelmed by the COVID-19 pandemic, and there was an evident absence of clear information and health policies. Ad hoc decisions that discriminated against older people, regardless of other factors, are also explicitly being made by health experts [36]. Indeed, older individuals are one of the most vulnerable segments of the population to the impact of the pandemic’s inevitable confinement [37].

Older adults pointed out the frequent postponement of medical specialist consultations during the pandemic. The pandemic has had a major impact on waits for elective surgeries, where many scheduled interventions are being canceled or postponed [38]. With the main focus being preventing the spread of the virus, many health services were negatively affected, including consultations and their rescheduling. Such recommendations also included the cancellations of non-urgent procedures, visits, and regular examinations for older people. Hence, older people are at greater risk for further mental and physical health deteriorations [32]. The withdrawal of elective interventions associated with the pandemic has driven an urgent need for provinces to reassess how health units manage their waiting lists [32].

Several specialist consultations were frequently postponed during the pandemic. Several recommendations were made for cancer management, depending on the number of cases and available resources in a country. In fact, in the presence of many positive cases of COVID-19, surgical postponements were frequently recommended [39]. Several factors must be considered regarding the management of older cancer patients during the current pandemic, such as their frailty, increased risk of infection, decreased immunity, social isolation, and the potential of death if infection occurs [40].

However, the delay or avoidance of medical care might increase the morbidity and mortality risk associated with treatable and preventable health conditions and might contribute to a reported excess in deaths directly or indirectly related to COVID-19 [41].

At the beginning of the pandemic, older adults at increased risk of contracting severe COVID-19 may have felt a real fear of exposure to the virus. [42]. Avoidance of care, postponement of medical consultations, or the closing or limiting of face-to-face services occurred out of concern for exposure to the virus. Therefore, affordable telehealth or home medical care can be a good alternative, as even in the context of a pandemic, patients with emergency cases must receive treatment without delay [43].

In addition, cancellations and postponements are expected to give rise to an overload of essential surgeries, exams, and consultations, with this surge further burdening an already overloaded health-care system and further delaying routine care and follow-up [32].

Changes in accessing health services during the pandemic have been challenging for the mental health of older adults. Older adults have often felt anxious, distressed and depressed. Online consultations and the virtual exchange of medical information are associated with concerns such as the quality of medical service, privacy, confidentiality, and security [44]. Some patients have shown a concern for their privacy as they do not share their health condition with family members; however, information might be heard during an online consultation, thus breaching their privacy [45]. In addition, during an online consultation, people who are at home are visible to people on the video call [46]. Online sessions also limit the quality of service as physical medical exams or procedural techniques cannot be performed. In addition, the health professionals’ and mainly the older individuals’ unfamiliarity with technological systems is another critical drawback [45].

Non-attendance of face-to-face activities such as consultations and day care services may worsen the quality of mental health, cognition, and the functioning of the older population [45].

Moreover, pre-existing mental health conditions are also much more likely to contribute to an increase in the development of chronic illnesses over time. Managing and treating pre-existing mental disorders is just as important as managing and treating chronic illnesses in older people [47].

Participants experienced traumatic situations when accessing health services. During the pandemic, potentially discriminatory health-care practices have been one of the most troubling and traumatic responses for older adults. Older individuals were seen as the lowest priority for life-saving treatment, regardless of other factors, such as functional health [36]. During this phase of the pandemic, discrimination, prejudice, and stereotypes regarding older people can cause lower quality health care. Older age and the pandemic negatively affect the mental health of this population as they are undervalued, seen as a burden, and discriminated against [48].

Older adults have experienced an increase in alcohol and drug consumption. Health professionals such as psychiatrists, therapists, nurses, community workers, and general practitioners specializing in mental health have a duty to intervene when there are changes in behaviors related to the mental health of their patients and the general population [6].

Participants reported frequent frustrated behavior when accessing health services. There was a preference for face-to-face consultations by some patients, who were also opposed to online consultations [49]. This was also due to poor video and audio quality and the constant loss of connections due to low internet speeds, making older adults feel more frustrated [49].

In addition, the lack of social presence and access to body language were seen as frustrating during online consultations [6]. It is important to note that much communication (about 80%) is non-verbal, which includes eye contact, patient posture, and hand movement, and these elements are difficult to see during online conversations [6].

Some participants reported emotional explosions due to difficulties, repetitive frustration, and fear when accessing health services. There is evidently a greater risk of psychological burden due to the increased severity of COVID-19, which is independent of several other significantly influential variables that describe stressors in the daily life of older adults during the pandemic. The results indicate the importance of multiple factors that have significantly affected psychological conditions during the past year. These results illustrate the dilemma that infection and illness in the social circle, as well as their countermeasures (social distancing), have negative consequences for our mental health [50].

Older adults experienced shifts in their sleeping and eating habits, mainly due to fear, anxiety, and depressive symptoms. Inevitably, the pandemic caused individuals to go through situations that require adaptability to new roles. For example, chronic illnesses can threaten older adults’ well-being and also motivate them to seek out, or perhaps reaffirm, their purpose in life as a way to adapt to and overcome such illnesses [50].

This study contains a number of limitations. A purposeful sample of participants was recruited. In addition, while a diverse sample of older adults was recruited with respect to nationality, the majority were Caucasian (87%). It would be necessary to analyze older adults from other ethnicities to assess the impact of the COVID-19 pandemic on these groups. Moreover, most of the participants were relatively healthy, were not living alone, and were living in relatively developed countries (with regular health systems, access to these health systems, and access to telemedicine). In addition, all participants possessed a basic knowledge of how to use new technologies; however, not all older people possess such knowledge or access [51]. In this context, the findings of this study therefore cannot be generalized and are not representative of the older population, although they can enrich future studies.

Notwithstanding these limitations, this study is innovative in addressing the important links between the COVID-19 pandemic, older adults’ access to health services, and mental health, considerations which bode further empirical attention. It is an extremely relevant topic, particularly during the COVID-19 pandemic. The difficulties that older adults can experience when accessing health services and their influence on mental health must be addressed now as COVID-19 continues to linger and also when planning for a future pandemic.

Within the context of COVID-19, technology is an extremely relevant concept as it gives all of us an opportunity to stay connected with each other. However, there are challenges for some groups, such as those shared by the older people taking part in this study. Older adults need greater knowledge as well as therapeutic and emotional support in this area [52].

The COVID-19 pandemic has impacted all groups in society in diverse ways, paving the way for more innovative perspectives in different areas, notwithstanding older adults’ access to health services. Telehealth and other virtual services have been growing continuously, and any such technologies must adapt to the needs and capabilities of older adults [53]. Adequate paths for the delivery of mental health care and strategies to maintain essential mental health services during any emergencies and beyond are essential [54].

Despite all the barriers and challenges that the COVID-19 pandemic has brought, many opportunities to improve the mental health of the older community have emerged: the increased use of technology by older adults, increased connectedness to family and friends, an increased quality of life for older individuals, reductions in social isolation, and better knowledge about the pandemic situation, among others [7,32,55,56,57]. Finally, the COVID-19 pandemic can also serve as an opportunity to address the shortage of professionals specializing in the area of aging. Among professionals from various disciplines, there may now be an increase in interest in working with seniors and with mental health issues [32]. Moreover, the findings of this study may contribute to theoretical implications for future gerontological policies by reporting older adults’ perspectives on health accessibility and its influence on mental health, which may facilitate older people’s access to resources, information, and services during future pandemics.

## 5. Conclusions

Population proportions of older adults across the globe continue to increase and the COVID-19 pandemic has affected all such groups. Thus, in this phase of public crisis, the effective promotion of health needs special attention. Future research should seek the perspectives of patients of various nationalities and also of health professionals. Both such groups can help us to learn more about the impact of the pandemic on access to health services, including those provided in nursing homes.

The older adults in our study perceived that the COVID-19 pandemic had an important influence upon their ability to access health services and their mental health. Indeed, most participants felt the influence of this health crisis in the increase in the physical inaccessibility of health-care providers; in the increased use of online health services and other virtual health care; in the growth of online medication prescription; in their difficulty in obtaining information and accessing non-communicable disease and mental health indicators; and in the postponement of medical specialist consultations. Moreover, they felt anxious, distressed, and depressive. They reported traumatic experiences and the consumption of substances of abuse. Moreover, they showed increased frustration, emotional outbursts, and shifts in their sleeping and eating habits.

The pandemic has also created new opportunities for the older population, such as in the area of mediation, consultations, and obtaining information, all of which are linked to the implementation of innovation in their lives.

Research indicates advances that enable us to provide excellent preventive and clinical care. Policies and interventions should take collective action so that we can promote mental health and prevent disease among the older population by focusing on providing equitable, person-centered, and high-quality care.

## Figures and Tables

**Table 1 ijerph-19-04277-t001:** Sample of sociodemographic and health characteristics.

Characteristics	Portuguese 145 (41.9)	Brazilian 110 (31.8)	English 91 (26.3)	Total 346 (100.0)
Age, Average ± SD				71.4 ± 4.63
Gender, *n* (%)				
Female	93 (64.1)	70 (63.6)	59 (64.8)	222 (64.2)
Male	52 (35.9)	40 (36.7)	32 (35.2)	124 (35.8)
Education, *n* (%)				
Primary school	81 (55.9)	62 (56.4)	51 (56.0)	194 (56.1)
Middle school	39 (26.9)	29 (26.4)	25 (27.5)	93 (26.8)
≥High school	25 (17.2)	19 (17.2)	15 (16.5)	59 (17.1)
Marital Status, *n* (%)				
Married or in a relationship	113 (77.9)	85 (77.3)	71 (78.0)	269 (77.7)
Not married or in a relationship	32 (22.1)	25 (22.7)	20 (22.0)	77 (22.2)
Household				
Live with someone	127 (87.6)	97 (88.2)	80 (87.9)	304 (87.9)
Live alone	18 (12.4)	13 (11.8)	11 (12.1)	42 (12.1)
Family Annual Income, *n* (%)				
≤25,000 €	75 (51.7)	57 (51.8)	48 (52.7)	180 (52.0)
>25,000 €	70 (48.3)	53 (48.2)	43 (47.3)	166 (48.0)
Perceived Health, *n* (%)				
Good	97 (66.9)	74 (67.3)	61 (67.0)	232 (67.1)
Poor	48 (33.1)	36 (32.7)	30 (33.0)	114 (32.9)

## Data Availability

The data presented in this study are available on request from the corresponding author. The data are not publicly available due to privacy and ethical restrictions.

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
