# Peer review of "Health Service Accessibility, Mental Health, and Changes in Behavior during the COVID-19 Pandemic: A Qualitative Study of Older Adults"

_ijerph, 2022, doi:10.3390/ijerph19074277_

Round 1

Reviewer 1 Report

It is the fact that Covid 19 pandemic has effected elderly patients in every way. This study is one of the important study showing how Covid pandemic  effects the elderly patients. I have a few suggestions for the article.
1- As stated in the limitations of the study, although the selected population was old, they could access and use technology. 
2- The group in which the study was conducted is relatively healthy (a group of patients without neurological or mental disease or sequelae) and not living alone (80% of them  live with others). 
3- The group in which the study was conducted has lived in relatively developed countries (with a regular health system, access to the health system, and infrastructure for telemedicine). Millions of elderly people around the world still do not have such opportunities.

The above items are the limitations of the study and it is not possible to generalize the study results.  However, it is important study to show the destruction of covid pandemi  in  elderly people and also this study may be a guide for  future studies.

4- The "conclusion" part of the article is long. It is recommended to narrow the "conclusion" section. Some parts in conclusion (especially parts where used references ) can be placed  in "discussion" section . 

5- The sentence on page 9 and in line 382-384 should be corrected.
6- The questions asked in the online questionnaire should be given as an appendix to the article. 
7- English language and style are need of minor spell check. 

Author Response

Sofia von Humboldt, PhD

William James Center for Research

ISPA – Instituto Universitário

Rua Jardim do Tabaco, 34, 1149-041 Lisboa, 

Portugal

Tel: 00351 9630 43947

[email protected]

Subject: Revision of manuscript for evaluation

Dear International Journal of Environmental Research and Public Health

 Editor-in-Chief

Paul B. Tchounwou

I would like to resubmit the revised version of the attached manuscript, "Health service accessibility, mental health and changes in behavior during the Covid-19 pandemic: A qualitative study with older adults” for consideration for publication in the International Journal of Environmental Research and Public Health journal.

Thank you for providing an appreciated feedback for our paper on the access to health services, mental health and changes of behavior during the Covid-19 pandemic in a cross-national older sample.

With the revision of this manuscript, I would like to describe in detail how the reviewer’ comments were addressed in the manuscript. Additionally, the changes in the manuscript are indicated in green color, to facilitate the Editor and the reviewers’ reading.

Reviewer #1:

Comment 1: It is the fact that Covid 19 pandemic has effected elderly patients in every way. This study is one of the important study showing how Covid pandemic effects the elderly patients. I have a few suggestions for the article.
1- As stated in the limitations of the study, although the selected population was old, they could access and use technology.

2- The group in which the study was conducted is relatively healthy (a group of patients without neurological or mental disease or sequelae) and not living alone (80% of them live with others). 

3- The group in which the study was conducted has lived in relatively developed countries (with a regular health system, access to the health system, and infrastructure for telemedicine). Millions of elderly people around the world still do not have such opportunities.

The above items are the limitations of the study and it is not possible to generalize the study results.  However, it is important study to show the destruction of covid pandemi  in  elderly people and also this study may be a guide for  future studies.

Response 1: Thank you for your analysis and theoretical and methodological positive comments. In relation to this comment, we reformulated the limitations section in the Discussion: “Moreover, most of the participants were relatively healthy, not living alone and were living in relatively developed countries (with a regular health system, access to the health system, and access to telemedicine). In addition, all participants had a basic knowledge of how to use new technologies; however, not all older people would have such knowledge or access [51]. In this context, the findings of this study therefore cannot be generalized however they can enrich future studies.”

Comment 2: The "conclusion" part of the article is long. It is recommended to narrow the "conclusion" section. Some parts in conclusion (especially parts where used references ) can be placed  in "discussion" section . 

Response 2:  Please go to the Discussion section, in which we narrowed the conclusion section and placed some parts in the  Discussion section.

Comment 3: The sentence on page 9 and in line 382-384 should be corrected.

Response 3: The above sentence was corrected.

Comment 4: The questions asked in the online questionnaire should be given as an appendix to the article. 

Response 4: The questions were included in an appendix to the article.

Comment 5: English language and style are need of minor spell check. 

Response 5: English language and style was checked. 

We very much appreciated your comments which helped us to improve our paper and make it more suitable and pertinent to the readers of International Journal of Environmental Research and Public Health.

We trust that this reviewed version of the manuscript addressed the reviewer’s concerns and that the contents of this paper are pertinent to the understanding of the access to health services, mental health and changes of behavior during the Covid-19 pandemic in a cross-national older sample.

Thank you for considering my revised manuscript.

Sincerely,

Sofia von Humboldt

Reviewer 2 Report

Dear authors,

This research paper describes the actual and innovative topic – Health service accessibility, mental health and changes in behavior during the Covid-19 pandemic: A qualitative study with older adults. Authors notice, that the objectives of the study are understanding the influence of the COVID pandemic on older adults’ access to health services; and exploring how health service accessibility during the pandemic influenced their mental health and self-reported changes of behavior. As well, authors point out,  that this study is innovative by addressing the important link between the Covid pandemic, older adults’ access to health services, and mental health, which bodes further empirical attention.

And I would like to share with authors some doubts and remarks too: it seems important to notice that the abstract is not very clear. Thus, abstract should be explained and described clearly and specifically. As well, it seems important to notice, that it would be needed to concentrate on the discussion and conclusions of the study. Thus, when developing sections of "Discussion" and "Conclusions" it would be needed to include to the debate more future oriented theoretical implications, thus accessing deeper discussion and concluding insights.

Author Response

Sofia von Humboldt, PhD

William James Center for Research

ISPA – Instituto Universitário

Rua Jardim do Tabaco, 34, 1149-041 Lisboa, 

Portugal

Tel: 00351 9630 43947

[email protected]

Subject: Revision of manuscript for evaluation

Dear International Journal of Environmental Research and Public Health

 Editor-in-Chief

Paul B. Tchounwou

I would like to resubmit the revised version of the attached manuscript, "Health service accessibility, mental health and changes in behavior during the Covid-19 pandemic: A qualitative study with older adults” for consideration for publication in the International Journal of Environmental Research and Public Health journal.

Thank you for providing an appreciated feedback for our paper on the access to health services, mental health and changes of behavior during the Covid-19 pandemic in a cross-national older sample.

With the revision of this manuscript, I would like to describe in detail how the reviewer’ comments were addressed in the manuscript. Additionally, the changes in the manuscript are indicated in green color, to facilitate the Editor and the reviewers’ reading.

Reviewer #2

Comment 1: This research paper describes the actual and innovative topic – Health service accessibility, mental health and changes in behavior during the Covid-19 pandemic: A qualitative study with older adults. Authors notice, that the objectives of the study are understanding the influence of the COVID pandemic on older adults’ access to health services; and exploring how health service accessibility during the pandemic influenced their mental health and self-reported changes of behavior. As well, authors point out,  that this study is innovative by addressing the important link between the Covid pandemic, older adults’ access to health services, and mental health, which bodes further empirical attention.

Response 1: We thank you for your constructive feedback on our manuscript.

Comment 2: And I would like to share with authors some doubts and remarks too: it seems important to notice that the abstract is not very clear. Thus, abstract should be explained and described clearly and specifically.

Response 2: We reformulated the abstract in order to be an objective representation of the article and followed the style of structured abstracts, but without headings: 1) Background; 2) Methods; 3) Results; and 4) Conclusion.

Comment 3: As well, it seems important to notice, that it would be needed to concentrate on the discussion and conclusions of the study. Thus, when developing sections of "Discussion" and "Conclusions" it would be needed to include to the debate more future oriented theoretical implications, thus accessing deeper discussion and concluding insights.

Response 3: Please go to the Discussion and  Conclusions sections, in which future oriented theoretical implications were debated.

We very much appreciated your comments which helped us to improve our paper and make it more suitable and pertinent to the readers of International Journal of Environmental Research and Public Health.

We trust that this reviewed version of the manuscript addressed the reviewer’s concerns and that the contents of this paper are pertinent to the understanding of the access to health services, mental health and changes of behavior during the Covid-19 pandemic in a cross-national older sample.

Thank you for considering my revised manuscript.

Sincerely,

Sofia von Humboldt

Reviewer 3 Report

Dear Authors,

please consider the following suggestions, listed point by point according to article's structure:

Introduction

-“To date there are no published studies exploring older adult access to health services 87 during the Covid-19 pandemic”: there are some studies published sicnce 2020, analysing older adults, Covid-19 and healthcare services, please consider what is presented in these studies that could support this study’s aims (ie.: "Ageism and COVID-19: what does our society's response say about us?", by Fraser S et alii);

Materials and Methods

-Please specify how were participants enrolled and how the sample was identified and/or calculated. What would the authors want this sample to be representative of?  Writing “A purposeful sample of participants was recruited.” in Discussion, in not enough. Moreover, please consider the lack of focus on technique in this sentence.

- If implemented, the methods section can be more informative, i.e. explaining the researchers’ choices in identifying themes and deepening methodology in general (instruments or softwares used, for example).

-Concerning participants’ characteristics, what about psychopathological and/or pathological medical history?

-The online questionnaire should be attached in Supplementary materials or cited in tables or figures in order to let the reader consider the specific questions used by researchers;

Results

-Please consider that reporting participants’ names or nicknames, unless this datum is needed for a comparison between respondents, appears unsustainable;

-Abouth themes, please explain the criteria for identifying “anxiety”, “depression”, “trauma”, “emotional explosion”;

-About the specific aim number2, please compare these study’s results with other studies;

Results

- Please consider that providing insights about methodology let researcher deepen results and consequently the results.

Author Response

Sofia von Humboldt, PhD

William James Center for Research

ISPA – Instituto Universitário

Rua Jardim do Tabaco, 34, 1149-041 Lisboa, 

Portugal

Tel: 00351 9630 43947

[email protected]

Subject: Revision of manuscript for evaluation

Dear International Journal of Environmental Research and Public Health

 Editor-in-Chief

Paul B. Tchounwou

I would like to resubmit the revised version of the attached manuscript, "Health service accessibility, mental health and changes in behavior during the Covid-19 pandemic: A qualitative study with older adults” for consideration for publication in the International Journal of Environmental Research and Public Health journal.

Thank you for providing an appreciated feedback for our paper on the access to health services, mental health and changes of behavior during the Covid-19 pandemic in a cross-national older sample.

With the revision of this manuscript, I would like to describe in detail how the reviewer’ comments were addressed in the manuscript. Additionally, the changes in the manuscript are indicated in green color, to facilitate the Editor and the reviewers’ reading. 

Reviewer #3

Comment 1: Dear Authors, please consider the following suggestions, listed point by point according to article's structure: Introduction: -“To date there are no published studies exploring older adult access to health services 87 during the Covid-19 pandemic”: there are some studies published sicnce 2020, analysing older adults, Covid-19 and healthcare services, please consider what is presented in these studies that could support this study’s aims (ie.: "Ageism and COVID-19: what does our society's response say about us?", by Fraser S et alii);

 Response 1: We thank you for your relevant comments. The above paragraph in Introduction section was modified in order to illustrate that few studies explored older adults in the context of healthcare services.

 Comment 2: Materials and Methods: -Please specify how were participants enrolled and how the sample was identified and/or calculated. What would the authors want this sample to be representative of?  Writing “A purposeful sample of participants was recruited.” in Discussion, in not enough. Moreover, please consider the lack of focus on technique in this sentence. - If implemented, the methods section can be more informative, i.e. explaining the researchers’ choices in identifying themes and deepening methodology in general (instruments or softwares used, for example).

Response 2: Please go to the Methods section, first paragraph which was reformulated in order to include the requested information: For this study, 373 people were initially contacted, and 367 agreed to participate. Recruiting methods were varied, namely, message boards, senior universities, community center list-serves, personal emails, and social networks ads. Twenty-one participants were excluded due to lack of availability or incomplete data provided, yielding an unweighted response rate of 92.8% and a weighted response rate of 93.8%.

The final purposeful sample included 346 participants from three nationalities (Brazilian, English, Portuguese), (64.2% women), aged between 65 and 79 years (M=71.5; SD=4.65). 77,7% participants were married or in a romantic relationship, 12.1% lived alone and 17.1% finished university (see Table 1). Moreover, in the Limitations section, it is indicated that results cannot be generalized and are not representative of the older population: “In this context, the findings of this study therefore cannot be generalized and cannot be representative of the older population, however they can enrich future studies.”

Furthermore in the discussion section, the paragraph indicating limitations was reformulated in order to clarify them: ”This study contains a number of limitations. A purposeful sample of participants was recruited. In addition, while a diverse sample of older adults was recruited with respect to nationality, the majority were Caucasian (87%). It would be necessary to analyze older adults from other ethnicities to assess the impact of the Covid-19 pandemic on these groups. Moreover, most of the participants were relatively healthy, not living alone and were living in relatively developed countries (with a regular health system, access to the health system, and access to telemedicine). In addition, all participants had a basic knowledge of how to use new technologies; however, not all older people would have such knowledge or access [51]. In this context, the findings of this study therefore cannot be generalized and cannot be representative of the older population, however they can enrich future studies.”

Finally the Data analysis section was reformulated in order to deepen the information about the choices of themes, software and instruments: “After having the sample information provided in the semi-structured interview protocol, a detailed analysis was performed to ensure exhaustive contact with the data. Data were analyzed, by applying content analysis [23].

First, for a more organized and complete analysis of the information, a codebook was created, wherein each important category was assigned a code [23]. All interviews were coded by three researchers independently, and all processes were considered reliable (k = 0.87). In this study, a p-value ≤ 0.05 was considered in all analyses.

The main themes and sub-themes were defined and then the categorization process began. At this stage, with the consent of the three psychologists, the main themes were grouped into clear and independent categories. These categories had small and intuitive names. In order to have a reliable and valid system, general principles of classification and categorization of qualitative data were taken into account: homogeneity (it was organized based on common elements); relevance (reflects the importance of the categories); and objectivity and fidelity (objective and trustworthy categories). These principles were maintained until the end of the process. The above process was handled manually.”

Comment 3: -Concerning participants’ characteristics, what about psychopathological and/or pathological medical history?

Response 3: We indicated in the study eligibility criteria in the Methods section that “all the participants did not have a history of problems with cognitive impairment functions: psychiatric or neurological illnesses, or history of drug or alcohol abuse, among others.”

Comment 4: -The online questionnaire should be attached in Supplementary materials or cited in tables or figures in order to let the reader consider the specific questions used by researchers;

Response 4: The questionnaire was included in an appendix to the article.

Comment 5: Results: -Please consider that reporting participants’ names or nicknames, unless this datum is needed for a comparison between respondents, appears unsustainable;

Response 5: In order to clarify all the quotes and pseudonyms, we included the following sentence in the Results section: “Key quotes representing the diversity of narratives of older people were selected to illustrate the diversity of the studied sample. All names are pseudonyms.” Moreover, we indicated in the methods section that “all information would be used only for this study and would be anonymous.”

Comment 6: Abouth themes, please explain the criteria for identifying “anxiety”, “depression”, “trauma”, “emotional explosion”;

Response 6: For the mental health topics, we used the American Psychiatric Association (2000). Diagnostic and statistical manual of mental disorders (4th ed., Text Revision). Washington, DC: Author.

Comment 7: About the specific aim number2, please compare these study’s results with other studies;

Response 7: Please go to the Discussion section, last paragraph, page 16 to first paragraph page 19, in which the authors compare the study results with other studies.

Comment 8: Results: Please consider that providing insights about methodology let researcher deepen results and consequently the results.

Response 8: Please go to the Methods section, which was reformulated in order to provide more insight and clarify the data collection and analysis.

We very much appreciated your comments which helped us to improve our paper and make it more suitable and pertinent to the readers of International Journal of Environmental Research and Public Health.

We trust that this reviewed version of the manuscript addressed the reviewer’s concerns and that the contents of this paper are pertinent to the understanding of the access to health services, mental health and changes of behavior during the Covid-19 pandemic in a cross-national older sample.

Thank you for considering my revised manuscript.

Sincerely,

Sofia von Humboldt